# Association of Lactase Persistence Genotypes (rs4988235) and Ethnicity with Dairy Intake in a Healthy U.S. Population

**DOI:** 10.3390/nu11081860

**Published:** 2019-08-10

**Authors:** Elizabeth L. Chin, Liping Huang, Yasmine Y. Bouzid, Catherine P. Kirschke, Blythe Durbin-Johnson, Lacey M. Baldiviez, Ellen L. Bonnel, Nancy L. Keim, Ian Korf, Charles B. Stephensen, Danielle G. Lemay

**Affiliations:** 1USDA ARS Western Human Nutrition and Research Center, Davis, CA 95616, USA; 2Genome Center, University of California Davis, Davis, CA 95616, USA; 3Department of Nutrition, University of California, Davis, CA 95616, USA; 4Division of Biostatistics, University of California Davis, Davis, CA 95616, USA

**Keywords:** lactase persistence, lactose intolerance, rs4988235, dairy, alternative plant-based milk, dietary recalls, healthy American population

## Abstract

Lactase persistence (LP) is a trait in which lactose can be digested throughout adulthood, while lactase non-persistence (LNP) can cause lactose intolerance and influence dairy consumption. One single nucleotide polymorphism (SNP ID: rs4988235) is often used as a predictor for dairy intake, since it is responsible for LP in people in European descent, and can occur in other ethnic groups. The objective of this study was to determine whether rs4988235 genotypes and ethnicity influence reported dairy consumption in the United States (U.S.). A food frequency questionnaire (FFQ) and multiple Automated Self-Administered 24-h recalls (ASA24^®^) were used to measure habitual and recent intake, respectively, of total dairy, cheese, cow’s milk, plant-based alternative milk, and yogurt in a multi-ethnic U.S. cohort genotyped for rs4988235. Within Caucasian subjects, LP individuals reported consuming more recent total dairy and habitual total cow’s milk intake. For subjects of all ethnicities, LP individuals consumed more cheese (FFQ *p =* 0.043, ASA24 *p =* 0.012) and recent total dairy (ASA24 *p =* 0.005). For both dietary assessments, Caucasians consumed more cheese than all non-Caucasians (FFQ *p =* 0.036, ASA24 *p*
*=* 0.002) independent of genotype, as well as more recent intake of yogurt (ASA24 *p*
*=* 0.042). LP subjects consumed more total cow’s milk than LNP, but only when accounting for whether subjects were Caucasian or not (FFQ *p*
*=* 0.015). Fluid milk and alternative plant-based milk consumption were not associated with genotypes or ethnicity. Our results show that both LP genotype and ethnicity influence the intake of some dairy products in a multi-ethnic U.S. cohort, but the ability of rs4988235 genotypes to predict intake may depend on ethnic background, the specific dairy product, and whether intake is reported on a habitual or recent basis. Therefore, ethnicity and the dietary assessment method should also be considered when determining the suitability of rs4988235 as a proxy for dairy intake.

## 1. Introduction

Lactose is the primary carbohydrate in cow’s milk and requires an enzyme called lactase (EC 3.2.1.108) for hydrolysis. Lactase is encoded by the *LCT* gene, which is dominantly expressed in the small intestine. While lactase activity is high during infancy, activity diminishes after weaning in about two-thirds of the population [1,2]. However, in some populations, lactase activity is retained into adulthood, and lactose can continue to be digested in individuals carrying a heritable trait known as lactase persistence (LP). The global distribution of the LP phenotype varies in different populations, with low frequencies in people of Asian descent and high (over 75%) frequencies in some populations of European and African descent [1]. Several single nucleotide polymorphisms (SNPs) are associated with LP and vary in frequencies in different populations [3,4,5]. One SNP (ID: rs4988235) occurs approximately 14 kb upstream of *LCT* and confers LP, where the dominant A allele is associated with LP, and the G allele with lactase non-persistence (LNP). rs4988235 is responsible for LP in European-descending populations but is one of several SNPs that can lead to LP in African-descending populations [4,5].

Unlike LP individuals, LNP persons have low lactase levels and may become lactose intolerant and may experience gastrointestinal discomfort following lactose consumption. Although lactose-intolerant persons may be able to consume moderate amounts of lactose without experiencing symptoms (up to approximately 15 g of lactose, or about 1 cup of milk), many who are lactose intolerant avoid dairy altogether [1,6]. Indeed, LNP individuals have been shown to consume less milk than LP individuals [7,8,9]. Fermented products such as cheese and yogurt typically have lower amounts of lactose compared to milk [10,11], and may be tolerated better than milk for the lactose intolerant. Some studies reported no differences in yogurt or cheese consumption between LP and LNP individuals [8,9,12,13], although a few have found some differences for yogurt [14,15]. Lactase-treated dairy products and plant-based alternative “milk” products, including almond, rice, and soy “milk” are possible substitutes for cow’s milk for lactose intolerant persons, since they are lactose-free. However, to our knowledge, no studies have been done to determine the association between plant-based alternative “milk” consumption and the LP genotype.

Other factors such as ethnic background may also influence diet and therefore dairy intake. Several analyses of the National Health and Nutrition Examination Survey (NHANES) data from different years have reported on differences in diet among non-Hispanic Whites, Mexican-Americans, and non-Hispanic Blacks in the U.S. For example, overall energy, saturated fat, and sugar intake differed among a U.S. population of non-Hispanic Whites, Mexican-Americans, and non-Hispanic Blacks [16]. With regards to dairy, Non-Hispanic Blacks consumed significantly less total dairy compared to non-Hispanic Whites and Mexican Americans under the age of 50 [17]. More non-Hispanic Whites met the Dietary Guidelines for Americans recommended intake for dairy compared to non-Hispanic Blacks and Mexican-Americans [18]. Indeed, the proportion of income spent on different food products varies among Caucasians, Asians, and African Americans [19]. Dairy is a key dietary source of important nutrients, including calcium, potassium, and vitamin D, and has been associated with bone health and reduced risk of diabetes, cardiovascular disease, and mortality in several observational studies [20,21,22,23]. Low consumption for some ethnicities may have an impact on health, especially among groups who are at higher risks for these diseases.

A few studies have investigated the association of the LP genotype with dairy intake in North American populations [7,15], but most focus on northern European populations, where dairy consumption is higher and the study populations are ethnically homogeneous [8,9,14,15,24,25]. Given the fact that the U.S. population is ethnically diverse, and that dairy intake has been associated with bone, cardiovascular, and other health benefits, we studied the association between rs4988235 genotypes and ethnicity with dairy intake. The proposed hypotheses of this study were: (1) a larger proportion of subjects with an rs4988235 LP genotype (AA or AG) will consume cow’s milk compared to those with the LNP genotype (GG), while a larger proportion of LNP subjects will consume alternative milk compared to LP subjects, (2) the LP genotypes will be associated with a higher intake of servings of dairy and dairy products compared to the LNP genotype and a lower intake of alternative milk, and (3) Caucasians will consume more total dairy and dairy products than non-Caucasians, regardless of rs4988235 genotype. We report findings for two types of dietary assessment methods that capture intake over different time periods to understand habitual (over the past year) and recent (24-h recalls) dairy intake in relation to rs4988235 genotypes and ethnicity.

## 2. Participants and Methods

### 2.1. Study Population

Participants were from the Nutritional Phenotyping Study, which is a cross-sectional observational trial conducted by the United States Department of Agriculture, Agriculture Research Service Western Human Nutrition Research Center located on the University of California, Davis campus in Davis, California (CA), United States (USA), as described previously [26]. Participants were recruited for the Nutritional Phenotyping Study, registered at clinicaltrials.gov (Identifier: NCT02367287). Study participants included healthy males and females aged 18 to 65 years and with a body mass index (BMI) between 18.5–45.0 kg/m^2^. Data were collected from participants during two study visits scheduled over a 10 to 14-day period. Anthropometry measurements were taken on Visit #1 and blood samples were collected on Visit #2 by a certified phlebotomist. Participants were trained to use Automated Self-Administered 24-h recalls (ASA24^®^) during Visit #1 and randomly prompted to complete three recalls (two weekdays and one weekend day) between Visit #1 and Visit #2. A Block 2014 Food Frequency Questionnaire (FFQ) was administered by interview with trained study personnel during Visit #2. Subjects were disqualified from the study if they were unwilling to consume a mixed macronutrient challenge meal that contained animal products (egg) during Visit #2. As a consequence, vegans were effectively excluded from this study. Participants considered for the current study (Figure 1) were those with genotype information available (*n* = 227) and dietary intake assessments of sufficient quality (see “Dietary Intake Assessments”). Ethnicity was self-reported by the subjects in a demographic questionnaire. Subjects were grouped as Caucasian (*n* = 128), Black or African-American (hereafter referred to as “African-American”, *n* = 13), Hispanic or Latino (hereafter referred to as “Hispanic”, *n* = 32), Asian (*n* = 29), Other (*n* = 13), or multi-racial (*n* = 12). The “Asian” group is comprised of subjects who responded as Asian, East Asian, South Asian, or Southeast Asian; and “Other” is comprised of subjects who responded as Other, Pacific Islander, Middle Eastern, Native American, or declined to respond. Multi-racial subjects (“Multi”), identified as more than one ethnic group. Subjects who identified as African-American, Asian, Caucasian, and Hispanic were included in any analyses including ethnicity (“single ethnicity” total *n* = 202, FFQ *n* = 198, ASA24 *n* = 195), and all ethnicities (including “Multi” and “Other”) were included for genotype-only analyses (*n* = 210, Figure 1).

### 2.2. Genomic DNA Purification and Genotyping

DNA was extracted from 8 mL of whole blood using a PAXgene^TM^ Blood DNA Kit (PreAnalytiX GmbH, Hombrechtikon, Switzerland). Genomic DNA was quantified using a NanoPhotometer™ P300 (Implen, Westlake Village, CA, USA) and diluted to a concentration at 25 µg/mL with sterilized double distilled water. A predesigned TaqMan^®^ SNP probe for the LP genotype (Assay ID: C_2104745_10; SNP ID: rs4988235) was purchased from ThermoFisher Scientific (Carlsbad, CA, USA). The context DNA sequence for the SNP probe is GAGGAGAGTTCCTTTGAGGCCAGGG[A/G]CTACATTATCTTATCTGTATTGCCA, where [A/G] is the transition substitution. TaqMan genotyping reactions were performed using a TaqMan SNP assay-based polymerase chain reaction (PCR) (ThermoFisher Scientific) in an Applied Biosystems™ QuantStudio™ 7 Flex Real-Time PCR System according to the manufacturer’s instructions. Fifty ng of genomic DNA was added into each PCR reaction along with 0.25 µL of the TaqMan SNP probe, 5 µL of 2× TaqMan Genotyping Master Mix, and 2.75 µL of sterilized double distilled water. Allelic discrimination assays were performed using QuantStudio™ Real-Time PCR software (ThermoFisher Scientific). All the ambiguous genotypes were repeated in independent PCR reactions.

### 2.3. Dietary Intake Assessments

Two dietary assessments were used in this study to estimate dairy intake. Habitual intake over the past year was assessed from the Block 2014 Food Frequency Questionnaire (FFQ) by NutritionQuest [27]. FFQs were included in analysis if the participant completed the questionnaire and if the daily kcal amount was within the 5th to 95th percentiles of kcal intakes from NHANES data for males (650–5700 kcal/day) and females (600–4400 kcal/day) (*n* = 218 (Figure 1)) [28]. Servings of total yogurt and total cheese, recorded as cup equivalents per day, were calculated from the standard Block FFQ output. Servings of total dairy and total cow’s milk (i.e., cow’s milk included in recipes, mixed dishes, and beverage form) were estimated from the standard Block FFQ output with modifications depending on reported primary milk type (“milktype”) and whether chocolate milk was consumed. The Block FFQ standard output for total cow’s milk (and therefore the output for total dairy) includes servings of soy milk, but does not include servings of chocolate milk.

Servings of chocolate milk and plant-based alternative milks (“milktype” was soy, rice, or almond) were converted to servings using the Food Pattern Equivalents database cup equivalents weight (1 cup equivalent (serving) = 245 g of milk) [29]. If the subject’s reported “milktype” was cow’s (whole, 2%, 1%, or skim, as well as “Don’t Drink”), then the servings of chocolate milk was added to the total cow’s milk and total dairy variables. The Block FFQ default is to report any consumed dairy as 2% cow’s milk if the subject’s “milktype” is “Don’t Drink” or is missing. Servings of soy milk (i.e., “milktype” was soy) were subtracted from the total cow’s milk and total dairy variables (Appendix A).

We also generated additional variables to represent the estimated intake of plant-based alternative milk and fluid cow’s milk (Appendix A). Grams of each product consumed as a beverage, with cereal, or with coffee/tea (but not in a mixed dish or recipe) were added to the alternative milk and fluid cow’s milk variables according to the subject’s “milktype” (alternative: soy, rice, or almond; fluid cow’s: whole, 2%, 1%, skim, “Don’t Drink”, or missing). Grams of chocolate cow’s milk or chocolate alternative milk were added to the fluid cow’s and alternative milk variables, respectively (Appendix A) [29].

Recent intake was estimated from the self-administered ASA24 24-h dietary recall from the National Cancer Institute [30]. Subjects were asked to complete a total of four 24-h dietary recalls: one training recall during Visit #1 and three recalls at home during the 10 to 14 days that separated the two study visits. Single recalls were excluded from analysis if they were incomplete or if the total kcal was not within the NHANES 5th to 95th percentiles. Subjects with at least three recalls available were used for analysis (*n* = 214 (Figure 1)). Of these subjects, 63 subjects used the ASA24-2014 version, and 151 used the ASA24-2016 version. Details of dairy foods that differed between ASA24-2014 and ASA24-2016 are in the Appendix A; no significant differences were found in the distribution or means of any of the dairy variables. Servings of total yogurt and total cheese, recorded as cup equivalents per day, were calculated from the standard ASA24 output. Servings of total dairy and total cow’s milk, which included servings of milk consumed as a beverage or used in recipes, were estimated after subtracting soy milk servings (Appendix A) [31]. We calculated servings of plant-based alternative milk reported in the ASA24 if a subject recorded consuming soy milk, almond milk, or rice beverage (Appendix A). Similarly, servings of fluid cow’s milk were estimated when subjects reported consuming whole, 2%, 1%, skim, chocolate, or lactose-reduced milk as a beverage or by adding to cereal, coffee, and/or tea (Appendix A). Servings per subject, averaged over the total number of recalls (three or four per subject), were used as input for analysis. Additional information regarding data cleaning of the ASA24 data and calculations for the fluid cow’s milk and alternative milk variables are available in the Appendix A.

### 2.4. Statistical Analysis

All the statistical analyses were performed in R v.3.5.1. Cohort characteristics were compared among genotypes using Kruskal–Wallis tests with pairwise Wilcoxon rank-sum tests using the Benjamini–Hochberg adjustment to correct for multiple comparisons for continuous variables, and chi-squared testing was used for categorical variables (sex). The prevalence of the AA, AG, and GG alleles among ethnicities for all subjects with reported single ethnicity (*n* = 202) was compared with a two-sided Fisher’s exact test. Pairwise post hoc testing of allelic frequencies between ethnicities was conducted using the Benjamini–Hochberg adjustment to correct for multiple comparisons. Correlations between the quantity of dairy reported and metadata, as well as between FFQ and ASA24 datasets, were assessed using Kendall’s τ due to the presence of zeros from subjects reporting zero servings and the non-normal distribution of the data. Association of dairy and categorical metadata was tested using Kruskal–Wallis tests.

The main effects of interest were the rs4988235 genotypes and ethnic background. Since the A allele is dominant and associated with LP, we tested all the rs4988235 genotypes (AA, AG, and GG) as well as the LP status (LP genotype: AA/AG and LNP genotype: GG). Since rs4988235 is prevalent in European-descending populations (Caucasians), we studied the effect of rs4988235 genotypes on Caucasian-only subjects as well as among all ethnicities. Just over half (56%) of the subjects were Caucasian, and the number of subjects for each of the other ethnicities was comparatively small (Table 1). Therefore, we tested ethnic background using all the ethnic groups as well as Caucasian versus non-Caucasian groups. Models included all the genotypes, ethnicity, or genotypes and ethnicity as predictors; or LP genotype, identifying as Caucasian, or LP genotype*Caucasian as predictors.

We tested models including only the main effects as well as models adjusted for age, sex, height, weight, and/or body mass index (BMI). Since height, weight, and BMI were correlated, only one was included in the model with the following priority: first, BMI; second, height; third, weight. Forwards and backwards stepwise selection of variables was conducted using *stepAIC*. Logistic regression was used to test whether the proportion of “consumers” or “non-consumers” of dairy products differed among genetic and/or ethnicity variables. A consumer was defined as a subject reporting >0 servings of cheese, >0 servings of alternative milk, and ≥0.0333 servings of yogurt (at least 1 cup equivalent per 30 days) in both the FFQ and ASA24. A consumer of cow’s milk reported >0 servings in the FFQ and >0.1 servings in the ASA24 (Appendix A).

Linear regression was used to test whether the amount (servings) of total dairy or dairy products consumed was influenced by genotypes or ethnic background. Data were transformed to improve normality. Model fit was confirmed by assessing residuals. When linear model residuals indicated poor fit, ordinal logistic regression was performed on rank transformed data using the function *polr* from the package *MASS*. Only ordinal logistic regression was used for analysis of the yogurt, alternative milk, and fluid cow’s milk variable. The distributions were heavily skewed; these data could not be transformed to achieve a normal distribution, and the inspection of residual plots from linear models indicated a poor fit. For both ordinal and linear regression models, pairwise comparisons were conducted using *emmeans* from the package *emmeans* v.1.2.3 with Tukey’s adjustment for multiple comparisons. Analyses were conducted on the absolute number of servings reported as well as servings per 1000-kcal for both habitual and recent intake (based on a subject’s reported average total kcal per day) to account for the influence of overall dietary intake [32,33]. For all analyses, α = 0.05. Plotted and reported values are for servings or servings/1000-kcal, but *p* values correspond to findings using transformed data.

## 3. Results

### 3.1. Cohort Characteristics

Cohort characteristics are summarized in Table 1. The average age of GG subjects was less than that of AG subjects (*p =* 0.017) but not AA subjects, and the height of GG subjects was less than both AA and AG subjects (*p =* 0.002 and 0.003, respectively) (Table 1). Total energy intake did not significantly differ among genotypes, whether for habitual or recent intake (*p* > 0.05, data not shown).

Only one Asian subject was an AG genotype, and no Asian or Hispanic subject was an AA genotype (Figure 2). Accordingly, the frequency of the AA and AG genotypes differed significantly between Caucasians and Asians (*p =* 2.69 × 10^−5^ and 5.44 × 10^−5^, respectively), as well as for the AA genotype between Caucasians and Hispanics (*p =* 1.58 × 10^−5^). The frequency of the GG genotype differed significantly between Caucasians and all other ethnicities (African American *p =* 4.32 × 10^−4^, Hispanic *p =* 7.68 × 10^−10^, Asian *p* = 3.67 × 10^−15^), and between Asians and African Americans (*p =* 3.88 × 10^−2^).

### 3.2. Comparison of Habitual (FFQ) and Recent (ASA24) Dairy Intake

Two different dietary instruments were used to measure intake. Four ASA24 recalls were administered in a 10 to 14-day span to capture recent intake. The Block FFQ was used to measure habitual intake over the previous year. After cleaning the dietary data (as described in Section 2.3
*Dietary Intake Assessments*), a total of 210 subjects had both FFQs and sufficient ASA24 recalls. Significant but weak correlations between the FFQ and ASA24 were found for servings of total dairy (τ = 0.358), cheese (τ = 0.315), total yogurt (τ = 0.430), total cow’s milk (τ = 0.409), fluid cow’s milk (τ = 0.464), and alternative milk (τ = 0.522). Using the FFQ and ASA24s, 91% and 93% of all subjects, respectively, reported consuming fewer than three servings of dairy per day, which is the recommended amount for adults by the Dietary Guidelines for Americans [34].

### 3.3. Association of Age, Sex, and BMI with Dairy Intake

Before incorporating genotype and ethnicity, the association of age, sex, BMI, height, and weight with dairy intake was investigated for all the subjects. Some weak but significant associations were found between subject metadata and the amount of dairy consumed for both habitual and recent intake (all τ < 0.25).

#### 3.3.1. Habitual Intake

Results from the FFQ were used as a measure of habitual intake. Servings of total dairy were positively correlated with weight and height. Servings of cheese per 1000-kcal was also correlated with weight (Appendix A). There were no associations of height, weight, or BMI with fluid milk for all subjects, either as servings or energy-normalized servings. However, among only the consumers (reporting >0 servings), servings of fluid milk were positively correlated with height (Appendix A). Habitual intake of yogurt servings per 1000-kcal was significantly but negatively correlated with height and weight (Appendix A). Thus, heavier participants consumed more total dairy, more cheese, and less yogurt; taller participants consumed more total dairy, more fluid milk (among consumers only), and less yogurt.

The only significant association of dairy intake and age was for yogurt. Among consumers of yogurt (reporting ≥0.033 servings), yogurt servings per 1000-kcal was negatively correlated with age (Appendix A). In other words, for the subset of participants who consumed yogurt, the younger participants consumed more.

Some significant associations of habitual dairy intake were found with sex. Men consumed more servings of total dairy (*p =* 0.034), total cow’s milk (*p =* 0.003), and fluid cow’s milk (*p =* 6.9 × 10^−4^) than women, but not cheese or yogurt (data not shown). When reporting on a per 1000-kcal basis, men consumed significantly more fluid milk than women (*p =* 0.002), and women consumed significantly more yogurt than men (*p =* 0.007).

#### 3.3.2. Recent Intake

Results averaged across multiple ASA24s in a 10 to 14-day span were used as a measure of recent intake. Servings of total dairy was positively associated with height, weight, and BMI; servings per 1000-kcal were associated with BMI and weight but not height (Appendix A). Servings of cheese were positively correlated with BMI, height, and weight, while energy-adjusted servings were only associated with BMI and weight (Appendix A). Alternative milk was negatively correlated with BMI. There were no significant associations between height, weight, and BMI with fluid cow’s milk and total cow’s milk for all the subjects. However, among only the consumers (fluid milk: reporting >0 servings, total cow’s milk: >0.1 servings), both were positively correlated with height (Appendix A). Consistent with habitual intake, heavier participants had a higher recent intake of total dairy and cheese, while taller participants had a higher recent intake of total dairy, cheese, and fluid milk (among consumers only).

Recent intake of cheese was positively correlated with age, but not when energy-adjusted. Unlike habitual intake, there was no association of recent intake of yogurt with age.

Using recent intake data, men consumed more servings of total dairy (*p =* 0.024), but there were no differences between sexes when reporting on a per 1000-kcal basis (data not shown). Contrary to habitual intake, there were also no significant differences between the sexes for total cow’s milk, fluid milk, or yogurt.

### 3.4. Genotypes and Reported Consumers of Products in Caucasians

We examined the association of rs4988235 allelic difference with reported habitual (via the FFQ) and recent (via ASA24s) dairy consumption within the Caucasian-only subjects (FFQ *n* = 126, ASA24 *n* = 125), since this was the largest single ethnic group within the cohort, and the LP phenotype is monogenic in populations of European descent via rs4988235 [5]. We first tested the association of genotypes with whether subjects consumed dairy or dairy products, and then tested the association of genotypes with the amount of dairy consumed for only consumers of each product as well as all subjects.

For both habitual and recent intake, a subject was considered a ‘consumer’ of a product if they reported >0 servings of total dairy, total cheese, fluid cow’s milk, and alternative milk; and ≥0.033 servings of yogurt. A ‘consumer’ of total cow’s milk, which includes milk from recipes, was defined as reporting >0 servings for habitual intake and >0.1 servings for recent intake. Among Caucasians, the proportion of subjects who reported consuming or not consuming total dairy was not significantly influenced by genotype for either habitual or recent intake. All the Caucasians were habitual consumers of total cow’s milk and cheese. Surprisingly, genotype did not influence whether or not Caucasian participants were consumers of fluid cow’s milk, alternative milk, or yogurt.

### 3.5. Genotype and Dairy Intake in Caucasians

#### 3.5.1. Habitual Intake

The association of genotype with the quantity of habitually consumed total dairy, cheese, milk, milk alternatives and yogurt, estimated using the FFQ, was investigated in the Caucasian subset of participants. LP Caucasians habitually consumed significantly more total cow’s milk than LNP Caucasians (servings *p =* 0.049, servings/1000-kcal, *p =* 0.084, Figure 3). In contrast to total cow’s milk, which includes milk in recipes, the quantity of habitually consumed total dairy, cheese, fluid cow’s milk, and yogurt did not significantly differ in relation to genotypes. In addition, the quantity of alternative milk consumed was not significantly different between genotypes.

#### 3.5.2. Recent Intake

For recent intake measured by the ASA24s, AG Caucasians consumed significantly more servings of total dairy than GG Caucasians (*p =* 0.047), and the difference was similar when comparing LP and LNP Caucasians (*p =* 0.058) for the adjusted models. However, there were no significant differences in the amount of total dairy/1000-kcal for Caucasian subjects. Recent intake of cheese, total cow’s milk, fluid cow’s milk or yogurt did not significantly differ in relation to genotype in Caucasian participants.

Regarding alternative milk, there were no significant differences in the quantity of recent consumption between genotypes in Caucasians. However, among only the reported consumers via the ASA24 recalls (*n* = 29), Caucasians with the AG genotype consumed more alternative milk than AA subjects (*p =* 0.036) and GG subjects (*p =* 0.020), although there was no association with the LP genotype. This significant finding may be due to stratification bias and the relatively small sample size of Caucasian-only consumers.

### 3.6. Genotypes, Ethnicity, and Proportion of Consumers

Although rs4988235 has a high prevalence in people of European descent, this SNP may also occur in other ethnicities (Figure 2) [3,5,35]. Since both LP and ethnic background may influence dairy intake, we therefore studied the association of rs4988235 and ethnicity with whether subjects were consumers or non-consumers of dairy or dairy products. See Section 3.4 for definitions of consumers and non-consumers.

All the subjects reported consuming dairy in both dietary assessments, and thus all subjects were dairy consumers. All the subjects reported habitual consumption of cheese (>0 servings reported) via the FFQ (Table 2). The proportion of subjects reporting recent consumption (via ASA24s) of cheese was influenced by ethnicity, but pairwise comparisons revealed no significant results; this was likely due to the overall small proportion of cheese non-consumers (only six subjects).

All the subjects reported habitual consumption of total cow’s milk (>0 servings reported), which includes milk in recipes, but not fluid cow’s milk or alternative milk (Table 2). Neither rs4988235 genotypes, ethnic background, nor the combination of the two influenced whether a subject was a consumer or non-consumer of total cow’s milk, fluid cow’s milk, and alternative milk for both habitual and recent intake.

Some subjects reported consuming no yogurt, either habitually (FFQ) or recently (ASA24s). The proportion of subjects who reported recent consumption of yogurt was not significantly different between genotypes, ethnicities, or a combination of the genotype and ethnicity. Taken together, genotype and/or ethnicity did not have a significant impact on whether or not a participant was a consumer or non-consumer of total dairy, dairy products, or alternative milk.

### 3.7. Genotypes, Ethnicity, and Servings of Dairy Consumed

#### 3.7.1. Habitual Intake

Using the FFQ as a measure of habitual intake, LP subjects consumed more servings of cheese than LNP subjects (*p =* 0.043, Table 3), but not more total dairy, total cow’s milk, fluid cow’s milk, yogurt, and alternative milk. There were also no significant relationships between genotype and habitual dairy intake of any type using energy-adjusted servings.

There were numerous significant relationships between habitual dairy intake and ethnicity. When reporting on a per 1000-kcal basis, Caucasians consumed more total dairy than non-Caucasians (*p =* 0.040), but there were no significant differences using the absolute value of reported total dairy (Table 4). Reported servings of cheese significantly differed by ethnicity, either as absolute servings (*p =* 0.002) or energy-adjusted servings (*p =* 0.000). Caucasians reported consuming more cheese servings than all the non-Caucasian subjects (*p =* 0.036). Asians reported consuming significantly fewer servings of cheese than all the other ethnic groups (Table 4). There were similar findings for energy-adjusted cheese intake. The association of ethnicity with absolute servings of yogurt was not quite significant (*p =* 0.052). However, when reporting on a per 1000-kcal basis, African-Americans consumed less yogurt than Caucasians (*p =* 0.010, Figure 4).

LP subjects reported consuming more total cow’s milk servings than LNP subjects, while accounting for whether subjects identified as Caucasian or not (*p =* 0.015, Figure 3). However, the LP genotype was not a significant predictor of total cow’s milk when tested alone (Table 3). Ethnic background was also not a significant predictor of total cow’s milk intake, and there were no significant interactions (Table 4).

Since there were non-consumers of fluid cow’s milk, alternative milk, and yogurt, it was next determined whether there is a significant difference in the amount consumed in relation to genotype and/or ethnicity only among the consumers. However, neither genotypes nor ethnic background influenced fluid cow’s milk, alternative milk, or yogurt consumption among consumers of each product (fluid >0, alternative >0, yogurt ≥0.033 reported; Appendix A).

#### 3.7.2. Recent Intake

Recent intake of total dairy, as measured by ASA24s, differed significantly in relation to genotypes. Lactase persistent subjects reported consuming significantly more servings (*p =* 0.005) and servings per 1000-kcal (*p =* 0.023) of total dairy than LNP subjects (Table 3). Similar findings were found for the adjusted models. Specifically, AG subjects consumed significantly more servings than GG subjects (*p =* 0.022). This association was significant for the adjusted model as well, but genotype did not significantly influence total dairy/1000-kcal.

Caucasian subjects reported consuming more servings of total dairy than all the non-Caucasian subjects (*p =* 0.005, Table 4). Specifically, Caucasians consumed more than Asians (*p =* 0.013). These findings were consistent for servings/1000-kcal and the adjusted models. Ethnic background and genotype were only significant predictors of total dairy intake when tested alone; there were no significant findings for either when testing the effects of both together, and there was also no interaction.

Recent intake of cheese also differed by both genotype and ethnicity. Lactase-persistent subjects reported recently consuming significantly more servings of cheese (*p =* 0.012) than LNP subjects (Table 3). Recent intake of cheese also differed by ethnicity (*p =* 0.000). Caucasians consumed significantly more cheese than all the non-Caucasians (*p* = 0.002) (Table 4). Specifically, Asians consumed less than Caucasians (*p* = 0.000) and Hispanics (*p* = 0.026).

Genotype and ethnic background were not significant predictors of recent consumption of fluid cow’s milk or alternative milk (Table 3 and Table 4). There were also no significant differences among only the consumers (fluid milk *n* = 94, alternative *n* = 48; Appendix A).

Recent intake of yogurt did not differ by genotype, but did differ by ethnicity. For all the subjects, Caucasians consumed more servings of yogurt than all the non-Caucasian subjects (*p =* 0.042, Table 4 and Figure 4), with a similar *p* value for the adjusted model. There were no significant differences in yogurt consumption in relation to ethnicity or genotype for only the consumers of yogurt (*n* = 79).

In summary, a higher habitual intake of total cheese and recent intake of both total dairy and cheese were associated with the LP genotype. Total cow’s milk consumption differed in relation to genotypes only when also considering whether subjects were Caucasian or not. Ethnicity influenced both habitual and recent intake of cheese and yogurt.

## 4. Discussion

### 4.1. Genotype and Ethnicity in Relation to Consumption of Dairy and Alternative Milk Products

Given the changing demographics in the U.S., the rise in popularity of alternative milk products, and the decline in fluid milk sales, it was not known whether consumer genotypes and/or ethnicity were associated with modern dairy consumption in the U.S. In populations of European descent, lactase persistence is monogenic and driven by rs4988235 [5]. In the current study, the relationships between genotype (rs4988235), ethnicity, and dairy consumption were examined in a cross-sectional observational study of healthy adults in California.

We hypothesized that a larger proportion of subjects consuming cow’s milk would have an LP genotype and that a larger proportion of alternative milk consumers would be of the LNP genotype. Milk contains about 12 g of lactose per serving [11], and eliminating or reducing dairy intake can reduce lactose intolerance symptoms. Surprisingly, in this study cohort, rs4988235 genotypes and ethnic background did not significantly influence whether or not subjects reported consuming or not consuming total cow’s milk, fluid cow’s milk, or alternative milk products. In other words, genotype and ethnic background did not influence whether or not a participant was a consumer of milk, although genotype and ethnic background did influence how much milk was consumed.

Unexpectedly, all the subjects reported consuming some amount of dairy. This may be explained by the exclusion of subjects who were unwilling to consume the egg-containing challenge meal during Visit #2, effectively removing vegans from this study. Additionally, because the total dairy variable includes any amount of dairy that was added to any food, subjects who typically avoid dairy may have small amounts of dairy calculated from mixed dishes and recipes.

We hypothesized that dairy consumption would be higher among lactase persisters and among Caucasians. Significant differences in total cow’s milk consumption were found between LP and LNP Caucasians, but there were no differences among all the non-Caucasian subjects (Figure 3). For the entire cohort, no differences in total cow’s milk consumption were found in relation to genotypes unless the Caucasian ethnicity was also considered (Figure 3). These results suggest that the Caucasian subjects were mainly driving the difference in total cow’s milk consumption between all the LP and LNP subjects. Other studies, most of which included ethnically homogeneous European populations, have used rs4988235 as a genetic predictor of dairy intake, and while many have found that milk or total dairy consumption is associated with the LP genotype [7,8,9,12,36], a few studies have found no relationship between rs4988235 and milk intake [13,37]. Similar to our study, others have also found no difference in yogurt consumption between LP and LNP individuals [7,8,9,13]. Yogurt is a fermented product that contains less lactose than milk [11], and may be tolerated better by those who are lactose intolerant. In contrast with other reports [8,9,12,13], we observed higher consumption of cheese for LP individuals compared to LNP on both a habitual and recent intake basis. However, these studies included European populations that consume overall more cheese compared to the U.S. [38], and may also consume different types of cheese, which can have variable amounts of lactose depending on the specific type.

We hypothesized that alternative milk consumption would be higher for LNP subjects compared to LP subjects, since the consumption of non-dairy alternative products in lieu of dairy products is one strategy for managing lactose intolerance. Yet, the LP genotype did not influence how much alternative milk was consumed for both habitual and recent intakes (Table 3). These results suggest that LNP individuals might be reducing dairy intake but not substituting with alternative milk beverages. Genotype also did not influence whether subjects were a consumer of alternative milk (reported >0 servings). Sales of plant-based alternative milks (e.g., almond, soy, and rice milks) have increased over the last decade [39]. The U.S. national average price of these beverages are more than that of cow’s milk, and higher-income households purchase significantly more alternative milks compared to lower-income households [40]. Additionally, consumer perceptions of non-dairy milk being environmentally-friendly or beliefs about animal welfare also contribute to purchasing habits [41], indicating that factors other than lactose intolerance influence alternative milk consumption.

Total cheese and yogurt consumption were influenced by ethnic background according to both dietary assessments (Table 4 and Figure 4). Total dairy intake was also predicted by ethnic background for recent intake as well as energy-adjusted habitual intake. There were no interactions between the LP genotypes and ethnic background; ethnicity was significant without taking genotype into account. In other words, ethnicity influenced the consumption of cheese, yogurt, and total dairy regardless of the prevalence of LP in the given population. A different analysis of a U.S. population also found lower consumption of total dairy, yogurt, and cheese for Mexican-Americans, non-Hispanic Blacks, and other minorities compared to non-Hispanic Whites [42]. Differences in dairy consumption among ethnicities may be explained by higher rates of perceived lactose intolerance. Compared to the general U.S. population, African-Americans reported higher rates of lactose intolerance (24% compared to 11%) despite only 19% of those identifying as intolerant having been clinically diagnosed [43]. Another study also found that non-Hispanic Blacks, as well as Hispanics, had higher rates of self-reported lactose intolerance compared to non-Hispanic Whites; calcium intake from dairy foods was significantly lower for those with self-perceived lactose intolerance for all races [44]. It has also been proposed that the gut microbiome may adapt to ferment lactose in LNP individuals [45]. Thus, other factors such as culturally-defined diets, perceptions of lactose intolerance, or microbial adaptations may influence dairy intake.

### 4.2. Limitations

In the current study, two methods of dietary assessments were used, with the FFQ and ASA24 capturing intake over different time periods (over the past year versus over a 10 to 14-day period). Each method has different biases and limitations [46]. The FFQ is better suited to capture episodically consumed foods, while the ASA24 collects more detailed data such as cooking methods. Although nutrient values obtained from the ASA24 system were more similar to recovery biomarkers compared to values from the FFQ [32], there can be high day-to-day variability between recalls [46]. FFQs are convenient and often used in nutritional studies, including many studying the relationship between dairy intake, LP genotypes, and health outcomes [7,14,20,24,36,47]. Importantly, each method has different strengths and limitations, and some findings were consistent between the two types of reporting. Cheese was consistently influenced by the LP genotype and ethnicity for both habitual and recent intake, indicating a robust link. Therefore, both genotype and ethnicity systemically influence cheese intake, no matter if it is on a recent or habitual basis.

Some noteworthy limitations of both the FFQ and ASA24 systems were that they were limited in the number of ethnic food options, meaning that these dietary analyses may not completely capture a subject’s diet. Additionally, while the ASA24 system included lactose-reduced milk as an option, only one subject in the study population reported consuming lactose-reduced milk. According to the ASA24, no subjects reported consuming lactase enzyme or ultrafiltered milk, which has less lactose than conventionally processed milk, on the days that were recalled. The FFQ did not include lactose-reduced milk or ultrafiltered milk as options to choose for the main milk type. Since we assume that the reported servings of dairy contain lactose, we may thus be underestimating the differences in lactose-containing dairy consumption among different genotypes. Additionally, lactose intolerance was not clinically tested. Household income and consumer food choice motives were not considered in these analyses, and would be interesting to include in future analyses, as these may help explain the consumption habits of alternative milk products.

Another limitation of the study was the use of a single SNP, rs4988235, which confers lactase persistence in populations of European descent [5]. Multiple SNPs, including rs4988235, may confer LP in populations of African descent [3,4]. The convergent, polygenic evolution of lactase persistence in East African populations suggests that variants other than rs4988235 may influence milk consumption [48]. In this study, all three genotypes of rs4988235 were represented in the Caucasian and African-American subjects, although the majority of African-American subjects were of the GG genotype (Table 1). Similarly, a U.S. population with African ancestry from the 1000 Genomes Project was predominately of the GG genotype (70.5%), with lower prevalence of the AG and AA genotypes (24.6% and 4.9%, respectively) [35,49]. In this study, the majority of Hispanic subjects were of the GG genotype, with few subjects of the AG genotype and none of the AA genotype (Table 1). Another California-based population with Mexican ancestry was also predominantly GG (57.8%) and AG (35.9%), but 6.2% were AA [35]. The Asian subjects in this study were predominantly GG with only one AG Asian subject, and included those identifying as Asian, East Asian, Southeast Asian, and/or South Asian. All of the East Asian populations in the 1000 Genomes project were only carriers for the G allele. For a Southeast Asian (Gujarati Indian) population from Texas, the GG genotype was also the most prevalent (72.8%), although some were of the AG and AA genotypes (26.2% and 1%, respectively) [35]. Thus, the distribution of genotypes in our cohort are consistent with reports from the 1000 Genomes Project. However, we cannot exclude the possibility that SNPs other than rs4988235 confer lactase persistence in this cohort, particularly among the African American participants.

Lastly, although this study was open to enrolling subjects of all ethnicities, more than 50% of the cohort identified as Caucasian (Table 1). As such, the investigation of minority ethnicities was challenging due to the relatively small numbers per group. The number of participants of African descent, who may express other variants that confer lactase persistence, was particularly low (*n* = 13). Populations of Native American ancestry, who were excluded from dietary analysis in this study due to low numbers (“Other” ethnicity), had high proportions of the LNP genotype (GG, 68.2%) and low dairy consumption in another study [50]. Participants of Middle Eastern descent were also included in “Other” ethnicity group; a study of seven Iranian populations found LP in 0 to 29.9% of the groups with significant associations between rs4988235, which is another LP-related SNP (ID: rs182549), and the LP phenotype [51]. Due to the small number of individuals in some of the ethnic groups, the current study was likely underpowered for comparisons with individual ethnic groups. Nevertheless, there were statistically significant differences among some ethnicities for cheese and yogurt. Greater proportions of these populations in future studies would be valuable to further identify the relationship between ethnicity, LP genotypes, and dairy consumption.

### 4.3. Health Implications

The Dietary Guidelines for Americans recommend three servings of dairy per day for adults [52]. In our cohort, less than 10% of individuals reported consuming the recommended servings for both habitual and recent dietary analyses. The Healthy Eating Index (HEI) informs of how well a diet conforms to the Dietary Guidelines for Americans, with a higher score indicating better alignment to the guidelines. The HEI requires a daily intake of ≥1.3 servings of dairy per 1000-kcal to achieve a maximum score for the HEI dairy component [53]. The HEI scoring for each dietary component is based on energy-adjusted values so that diet quality, not quantity, can be assessed [33]. In our study, only 11% and 6% met the HEI standard for habitual and recent intakes, respectively. The recommendations set forth by the dietary guidelines and thus the HEI are to guide healthy eating patterns. Indeed, dairy is a major dietary source of calcium, vitamin D, and potassium, and consumption has been inversely associated with Type 2 diabetes [21], cardiovascular disease and mortality [20], osteoporosis [23], and metabolic syndrome [42,54]. Many of these disproportionately affect minorities [55,56,57], and low consumption of dairy for at-risk populations could have health implications. For example, calcium is associated with higher bone mineral density (BMD), and consumption of dairy has been linked with bone health [58]. Compared to African Americans, Caucasians, and Hispanics, Asian women have the lowest BMD [59], yet are consuming some of the lowest amounts of dairy (Table 3). Basically, overall, few individuals are achieving the recommended servings of dairy (>3 servings or ≥1.3 servings/1000-kcal), with some groups further avoiding consumption either due to lactase non-persistence/lactose intolerance (which may be perceived or clinical) or due to cultural influences. Both the LP genotype and ethnicity separately influenced dairy intake and should be considered when addressing the role of diet in nutrition and health. Consuming more yogurt in lieu of cow’s milk may be one way to achieve the recommended servings of dairy for LNP individuals, since yogurt has less lactose than cow’s milk [11]. Cheese may also be a suitable way to achieve the recommended servings, although the amount of lactose can be highly variable between different cheese types. However, overcoming low dairy consumption for some ethnic groups may be challenging. For example, introducing high quantities of yogurt into the diet may be difficult when yogurt and/or dairy are not part of a culture’s typical diet [60,61].

### 4.4. Implications for rs4988235 as a Proxy for Dairy Consumption

The LP SNP, rs4988235, has previously been used as a genetic proxy for milk and dairy consumption [7,12,13,14,21]. Our study results show a weak relationship between this SNP and dairy consumption for a multi-ethnic U.S. cohort. Dairy consumption was overall low, with little change in habitual consumption in relation to the LP genotype for the subjects of all ethnicities. The exception was for reported consumption of cheese, in which intake was influenced by genotype on both a habitual and recent basis for all subjects, even though there were no significant differences in consumption between LP and LNP among only the Caucasian subjects. Furthermore, even among Caucasians in the current study, the relationship between LP genetics and milk consumption was not particularly strong, having no influence over whether or not a participant was a consumer of cow’s milk. These results suggest that the rs4988235 SNP is not a strong proxy for dairy consumption in a U.S. population.

## 5. Conclusions

In a multi-ethnic cohort of healthy U.S. adults, the LP genotype did not influence whether a participant was a consumer of milk, but did influence the amount of milk consumed among Caucasians. Alternative milk consumption was not associated with LP genetics or ethnicity, suggesting that factors other than genotype and ethnicity have influenced alternative milk’s rising popularity. Within Caucasian subjects, the LP subjects reported consuming more recent total dairy and habitual total cow’s milk intake.

The LP genotype was associated with higher habitual intake of total cheese and recent intake of both total dairy and cheese. Ethnicity influenced the consumption of total dairy, cheese, and yogurt, regardless of the prevalence of LP in the given population, with Caucasians consuming more. Less than 10% of the entire cohort reported consuming the recommended servings of dairy, which can contribute to a shortfall of key nutrients such as calcium, vitamin D, and potassium. The deficit is magnified for non-Caucasian consumers, who report a lower consumption of dairy both habitually and recently. Considering that dairy is purported to be protective against some chronic diseases that disproportionately affect minorities, this suggests an opportunity to increase dairy consumption among all non-Caucasians, regardless of genotype, in order to meet requirements and positively influence health. Finally, LP genotypes, ethnicity, and dietary assessment methods influence reported dairy intake, and should be considered when determining the suitability of rs4988235 as a proxy for dairy intake.

## Figures and Tables

**Figure 1 nutrients-11-01860-f001:**
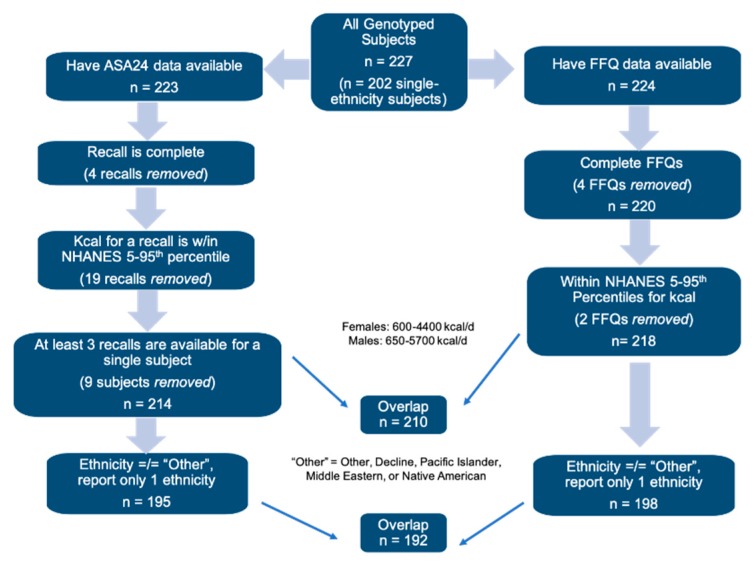
Flow chart for subject inclusion criteria for food frequency questionnaire (FFQ) or Automated Self-Administered 24-h recall system (ASA24 ^®^) analyses.

**Figure 2 nutrients-11-01860-f002:**
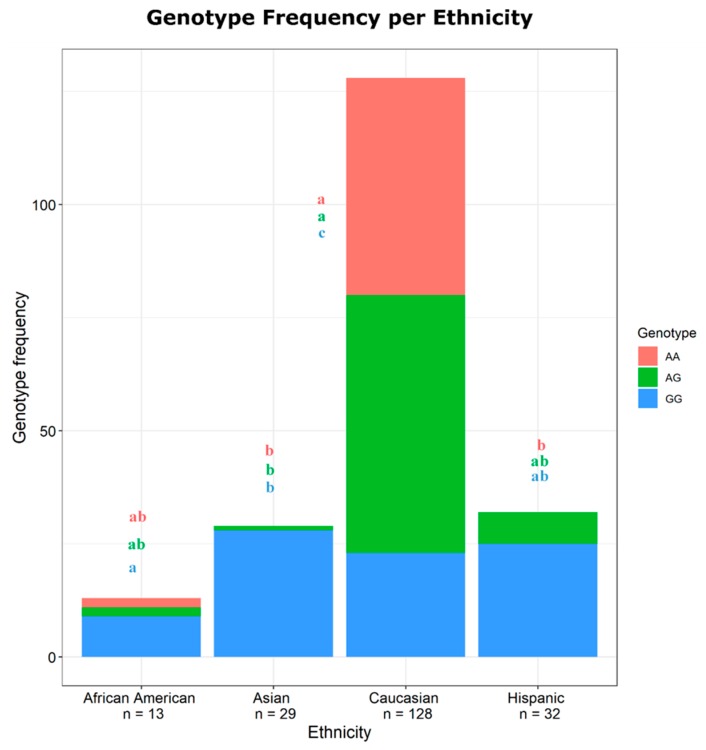
Genotype frequencies among ethnicities were compared with a Fisher’s exact test. Pairwise testing between ethnicities were conducted using the Benjamini–Hochberg adjustment for multiple comparisons. Ethnicities with different letters have significantly different frequencies for the AA (red), AG (green), or GG (blue) genotypes (*p* < 0.05).

**Figure 3 nutrients-11-01860-f003:**
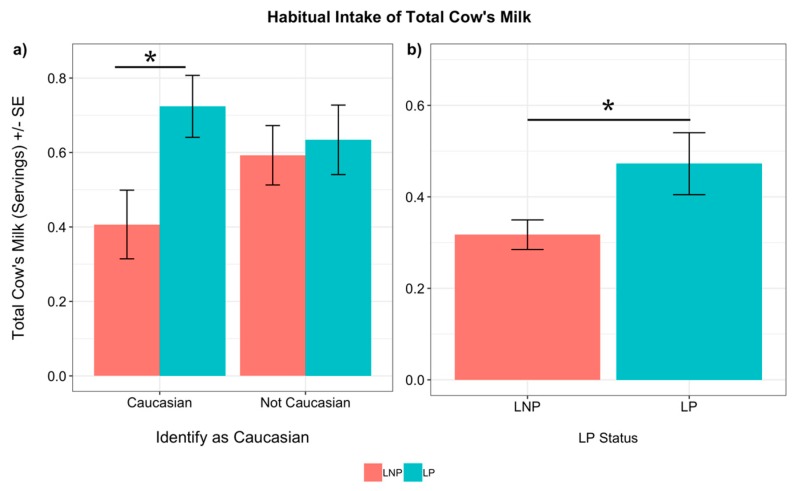
Reported habitual intake of total cow’s milk via the FFQ: (**a**) Unadjusted mean servings for lactase persistent (LP) and lactase non-persistent (LNP) subjects stratified by whether subjects identified as Caucasian (*n* = 126) or not (African-American, Asian, and Hispanic subjects; *n* = 72) and (**b**) Adjusted mean servings for LP (*n* = 114) and LNP individuals (*n* = 84), adjusted for whether subjects were Caucasian or not Caucasian. Habitual cow’s milk was not associated with the LP genotype unless tested in combination with the Caucasian ethnicity. *p* values were calculated from linear models using *log*-transformed values. Significant differences are denoted by asterisks (* *p* < 0.05). Error bars indicate standard error.

**Figure 4 nutrients-11-01860-f004:**
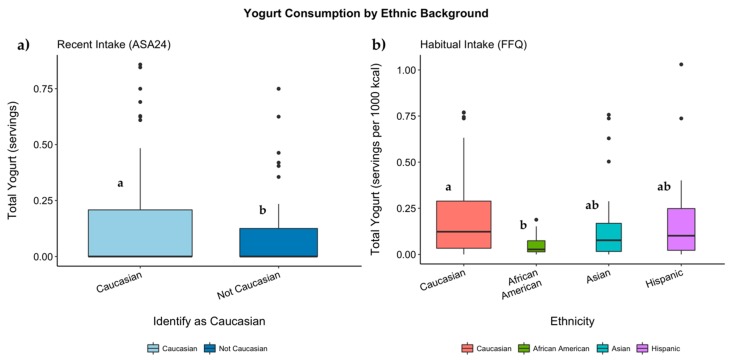
Boxplot of all subjects for yogurt for (**a**) recent intake measured via the ASA24 recalls and for (**b**) habitual intake measured via the FFQ. Box width is scaled to the number of subjects. *p* values were calculated using ordinal logistic regression. Adjustment for multiple comparisons was performed using the Tukey adjustment. Groups without a common letter differ significantly (*p* < 0.05).

**Table 1 nutrients-11-01860-t001:** Characteristics for all study subjects, as well as for each genotype, and subjects included for FFQ and ASA24 analyses.

		All Subjects (*n* = 227)	FFQ Subjects (*n* = 218)	ASA24 Subjects (*n* = 214)	rs4988235 Genotype (All Genotyped Subjects, *n* = 227)			
					AA (*n* = 53)	AG (*n* = 75)	GG (*n* = 99)	*p* ^1^
**Metadata mean (SD) or *n* (%)**	**Age (years)**	39.17 (13.96)	39.09 (13.73)	39.12 (13.82)	41.77 (13.67) ^a,b^	41.07 (12.91) ^a^	36.33 (14.49) ^b^	0.010
	**BMI kg/m^2^**	26.36 (4.65)	26.86 (4.63)	26.74 (4.60)	26.66 (5.42)	26.12 (4.10)	26.37 (4.63)	0.930
	**Height (cm)**	168.86 (9.91)	169.21 (9.77)	168.84 (9.73)	171.72 (9.24) ^a^	170.85 (9.87) ^a^	165.82 (9.52) ^b^	0.000
	**Weight (kg)**	77.04 (17.12)	77.33 (17.26)	76.62 (17.00)	80.82 (21.14)	77.95 (14.67)	74.32 (16.14)	0.113
	**Males**	94 (41.41%)	92 (42.20%)	87 (40.65%)	23 (43.40%)	32 (42.67%)	39 (39.39%)	0.860
**Ethnicity *n* (%)**	**African American**	13 (5.73%)	13 (5.96%)	11 (5.14%)	2 (3.77%)	2 (2.67%)	9 (9.09%)	0.010
	**Asian**	29 (12.78%)	27 (12.39%)	27 (12.62%)	0 (0.00%)	1 (1.33%)	28 (28.28%)	-
	**Caucasian**	128 (56.39%)	126 (57.80%)	125 (58.41%)	48 (90.57%)	57 (76.00%)	23 (23.23%)	-
	**Hispanic**	32 (14.10%)	32 (14.68%)	32 (14.95%)	0 (0.00%)	7 (9.33%)	25 (25.25%)	-
	**Other**	13 (5.73%)	11 (5.05%)	11 (5.14%)	0 (0.00%)	4 (5.33%)	9 (9.09%)	-
	**Multi**	12 (5.29%)	9 (4.13%)	8 (3.74%)	3 (5.66%)	4 (5.33%)	5 (5.05%)	-
**Genotype *n* (%)**	**AA**	53 (23.35%)	52 (23.85%)	51 (23.83%)	-	-	-	-
	**AG**	75 (33.04%)	71 (32.57%)	72 (33.64%)	-	-	-	-
	**GG**	99 (43.61%)	95 (43.58%)	91 (42.52%)	-	-	-	-

^1^ For the continuous variables, *p* values were calculated using a Kruskal–Wallis test with pairwise Wilcoxon rank-sum tests using the Benjamini–Hochberg adjustment for multiple comparisons. For the categorical variable (sex), the *p* value was calculated from a chi-squared test. For each genotype, groups without a common superscript letter differ (*p* < 0.05). BMI: body mass index.

**Table 2 nutrients-11-01860-t002:** The number of consumers (*n* (%)) per genotype and ethnicity for each product ^1^.

	Genotype			Ethnicity ^2^					
	AA	AG	GG	African-American	Asian	Caucasian	Hispanic	Multi	Other
**Habitual Intake (FFQ, *n* = 218)**									
Total Dairy (*n* = 218)	52 (100%)	71 (100%)	95 (100%)	13 (100%)	27 (100%)	126 (100%)	32 (100%)	9 (100%)	11 (100%)
Total Cheese (*n* = 218)	52 (100%)	71 (100%)	95 (100%)	13 (100%)	27 (100%)	126 (100%)	32 (100%)	9 (100%)	11 (100%)
Total Yogurt (*n* = 157)	43 (83%)	49 (69%)	65 (68%)	5 (38%)	19 (70%)	95 (75%)	23 (72%)	7 (78%)	8 (73%)
Total Cow’s Milk (*n* = 218)	52 (100%)	71 (100%)	95 (100%)	13 (100%)	27 (100%)	126 (100%)	32 (100%)	9 (100%)	11 (100%)
Fluid Cow’s Milk (*n* = 145)	38 (73%)	44 (62%)	63 (66%)	8 (62%)	19 (70%)	80 (63%)	22 (69%)	7 (78%)	9 (82%)
Alternative Milk (*n* = 58)	13 (25%)	22 (31%)	23 (24%)	5 (38%)	5 (19%)	36 (29%)	8 (25%)	2 (22%)	2 (18%)
**Recent Intake (ASA24, *n* = 214)**									
Total Dairy (*n* = 214)	51 (100%)	72 (100%)	91 (100%)	11 (100%)	27 (100%)	125 (100%)	32 (100%)	8 (100%)	11 (100%)
Total Cheese (*n* = 202)	50 (98%)	70 (97%)	82 (90%)	9 (82%)	23 (85%)	121 (97%)	31 (97%)	8 (100%)	10 (91%)
Total Yogurt (*n* = 79)	21 (41%)	27 (38%)	31 (34%)	2 (18%)	8 (30%)	51 (41%)	9 (28%)	4 (50%)	5 (45%)
Total Cow’s Milk (*n* = 174)	44 (86%)	60 (83%)	70 (77%)	9 (82%)	21 (78%)	103 (82%)	26 (81%)	6 (75%)	9 (82%)
Fluid Cow’s Milk (*n* = 94)	24 (47%)	34 (47%)	36 (40%)	4 (36%)	14 (52%)	61 (49%)	10 (31%)	2 (25%)	3 (27%)
Alternative Milk (*n* = 48)	11 (22%)	18 (25%)	19 (21%)	2 (18%)	3 (11%)	29 (23%)	9 (28%)	2 (25%)	3 (27%)

^1^ The total number of consumers are in parentheses following each product name. ^2^ Subjects who identified as “Other” and “Multi-ethnic” were not included in ethnicity analyses, but were included for genotype-only analyses.

**Table 3 nutrients-11-01860-t003:** Mean (SD) of reported servings and servings/1000-kcal of each product for all subjects by genotype.

	Habitual Intake (FFQ)					Recent Intake (ASA24)				
	AA	AG	GG	*p* ^1^	*p* ^2^	AA	AG	GG	*p* ^1^	*p* ^2^
**Servings·day**										
Total Dairy	1.86 (1.06)	1.68 (1.07)	1.62 (1.01)	0.279	0.196	1.57 (1) ^a,b^	1.64 (1) ^a^	1.25 (0.76) ^b^	**0.019**	**0.005**
Total Cheese	0.87 (0.50)	0.80 (0.47)	0.74 (0.54)	0.105	**0.043**	0.81 (0.60) ^a,b^	0.91 (0.69) ^a^	0.66 (0.56) ^b^	**0.030**	**0.012**
Total Yogurt	0.19 (0.21)	0.17 (0.18)	0.20 (0.30)	0.661	0.715	0.14 (0.21)	0.10 (0.16)	0.11 (0.20)	0.317	0.262
Total Milk	0.74 (0.84)	0.67 (0.75)	0.62 (0.69)	0.632	0.402	0.58 (0.76)	0.58 (0.54)	0.43 (0.49)	0.117	0.056
Fluid Milk	0.37 (0.72)	0.35 (0.66)	0.30 (0.52)	0.841	0.809	0.33 (0.73)	0.27 (0.42)	0.22 (0.45)	0.479	0.236
Alt. Milk	0.08 (0.20)	0.13 (0.45)	0.09 (0.27)	0.621	0.516	0.07 (0.19)	0.23 (0.64)	0.11 (0.29)	0.656	0.636
**Servings/1000-kcal·day**										
Total Dairy	0.93 (0.48)	0.85 (0.38)	0.81 (0.37)	0.297	0.250	0.73 (0.42)	0.74 (0.37)	0.62 (0.35)	0.075	**0.023**
Total Cheese	0.44 (0.23)	0.41 (0.20)	0.37 (0.22)	0.114	0.051	0.38 (0.26)	0.41 (0.29)	0.32 (0.25)	0.147	0.059
Total Yogurt	0.10 (0.10)	0.09 (0.11)	0.11 (0.20)	0.637	0.873	0.07 (0.10)	0.04 (0.07)	0.05 (0.10)	0.351	0.314
Total Milk	0.37 (0.42)	0.33 (0.31)	0.30 (0.30)	0.715	0.496	0.27 (0.32)	0.26 (0.24)	0.22 (0.25)	0.231	0.107
Fluid Milk	0.19 (0.40)	0.16 (0.27)	0.14 (0.22)	0.877	0.816	0.15 (0.32)	0.12 (0.18)	0.12 (0.24)	0.527	0.273
Alt. Milk	0.04 (0.11)	0.06 (0.19)	0.04 (0.13)	0.696	0.593	0.03 (0.08)	0.09 (0.25)	0.06 (0.17)	0.680	0.668

Within a genotype group, means without a common superscript letter differ (*p* < 0.05). *p* values were obtained from linear models of Box–Cox or log-transformed data or, if linear model residuals indicated a poor fit, then *p* values were obtained from ordinal logistic regression models. ^1^
*p* value for model with all rs4988235 genotypes as the predictor with the Tukey adjustment used to adjust for multiple comparisons. ^2^
*p* value for model with LP genotype as the predictor (LP = AA and AG; LNP = GG). *p* values in bold indicate significance at α = 0.05.

**Table 4 nutrients-11-01860-t004:** Mean (SD) of reported servings and servings/1000-kcal of each product for all subjects by ethnicity.

	Habitual Intake (FFQ)						Recent Intake (ASA24)					
	African-American	Asian	Caucasian	Hispanic	*p* ^1^	*p* ^2^	African-American	Asian	Caucasian	Hispanic	*p* ^1^	*p* ^2^
**Servings·day**												
Total Dairy	1.41 (0.41)	1.43 (1.08)	1.72 (1.06)	1.68 (0.87)	0.242	0.226	1.22 (0.91) ^a,b^	1.01 (0.62) ^a^	1.6 (0.98) ^b^	1.35 (0.76) ^a,b^	**0.014**	**0.005**
Total Cheese	0.91 (0.42) ^a^	0.53 (0.5) ^b^	0.82 (0.47) ^a^	0.79 (0.49) ^a^	**0.002**	**0.036**	0.73 (0.70) ^a,b^	0.39 (0.41) ^a^	0.85 (0.61) ^b^	0.76 (0.60^) b^	**0.000**	**0.002**
Total Yogurt	0.05 (0.06)	0.17 (0.22)	0.18 (0.19)	0.18 (0.22)	0.052	0.123	0.04 (0.09)	0.10 (0.19)	0.13 (0.20)	0.09 (0.19)	0.139	**0.042**
Total Milk	0.43 (0.21)	0.63 (0.75)	0.67 (0.79)	0.64 (0.53)	0.785	0.805	0.43 (0.45)	0.46 (0.45)	0.57 (0.64)	0.44 (0.58)	0.613	0.181
Fluid Milk	0.12 (0.16)	0.41 (0.67)	0.33 (0.68)	0.32 (0.47)	0.548	0.386	0.28 (0.47)	0.26 (0.39)	0.30 (0.57)	0.21 (0.58)	0.289	0.239
Alt. Milk	0.13 (0.24)	0.11 (0.37)	0.10 (0.36)	0.08 (0.27)	0.586	0.651	0.08 (0.18)	0.07 (0.21)	0.15 (0.50)	0.19 (0.40)	0.375	0.739
**Servings/1000-kcal·day**												
Total Dairy	0.69 (0.28)	0.73 (0.27)	0.9 (0.44)	0.78 (0.24)	0.153	**0.040**	0.58 (0.45)	0.53 (0.31)	0.74 (0.38)	0.66 (0.36)	**0.044**	**0.015**
Total Cheese	0.45 (0.25) ^a^	0.27 (0.18) ^b^	0.43 (0.23) ^a^	0.36 (0.11) ^a,b^	**0.000**	**0.004**	0.34 (0.31) ^a,b^	0.21 (0.19) ^a^	0.40 (0.27) ^b^	0.34 (0.22) ^b^	**0.006**	**0.009**
Total Yogurt	0.02 (0.03) ^a^	0.10 (0.13) ^a,b^	0.10 (0.10) ^b^	0.09 (0.10) ^a,b^	**0.019**	0.094	0.01 (0.03)	0.06 (0.11)	0.06 (0.09)	0.04 (0.10)	0.145	0.055
Total Milk	0.21 (0.14)	0.32 (0.27)	0.34 (0.37)	0.3 (0.23)	0.741	0.961	0.21 (0.21)	0.24 (0.24)	0.26 (0.27)	0.25 (0.34)	0.746	0.299
Fluid Milk	0.06 (0.10)	0.17 (0.24)	0.16 (0.33)	0.15 (0.22)	0.546	0.466	0.13 (0.21)	0.13 (0.20)	0.14 (0.24)	0.13 (0.31)	0.315	0.311
Alt. Milk	0.04 (0.07)	0.05 (0.13)	0.05 (0.16)	0.05 (0.18)	0.644	0.678	0.03 (0.08)	0.04 (0.13)	0.06 (0.19)	0.11 (0.26)	0.382	0.750

Within a group, means without a common superscript letter differ (*p* < 0.05). *p* values were obtained from linear models of Box-Cox or log-transformed data or, if linear model residuals indicated a poor fit, then *p* values were obtained from ordinal logistic regression models. ^1^
*p* value for model with ethnicity as the predictor with the Tukey adjustment used to adjust for multiple comparisons. ^2^
*p* value for model with whether subjects identified as Caucasian or not as the predictor. *p* values in bold indicate significance at α = 0.05.

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
