# Peer review of "Association of Lactase Persistence Genotypes (rs4988235) and Ethnicity with Dairy Intake in a Healthy U.S. Population"

_nutrients, 2019, doi:10.3390/nu11081860_

Round 1
Reviewer 1 Report
Association of lactase persistence genotypes (rs4988235) and ethnicity with dairy intake in a healthy U.S. population
Chin EL et al.
Lactose intolerance, whether perceived or actual, can lead to reduced dairy intake which may cause nutrient deficiencies. The stated objective of this manuscript was to study “the association of both rs4988235 and ethnicity with intake of total dairy as well as the association with the specific products cow’s milk, yogurt, cheese, and plant-based alternative milks.”
Key Points:
The manuscript lacks a clear hypothesis, which makes it hard to interpret the findings given they looked at the impact of multiple independent variables (genetics and ethnicity both alone and combined) on several dependent variables (total dairy, milk, cheese, yogurt, and alternative milks).
The bulk of the introduction is focused on the role of genetics in lactose intolerance. Thus, the inclusion of ethnicity in this study seems like an afterthought. Further, given that this gene is associated with LP/LNP in mostly Caucasians the rationale for looking at other ethnic groups was not fully clear nor understood.
Methods:
It is unclear to this review why both FFQ and 24-hour recall data were analyzed in the study. 24-hour recalls can be used to determine both daily and usual intakes via the NCI usual intakes method. Thus, it is unclear why the need for the FFQ when one set of data could be used if the authors wanted to look at both usual and daily intake.
The type of dietary assessment method used directly impacted the results, thus confounding interpretation of the findings. For example, in Table 3 there are several instances where results across genetic groups are significant for one method but not the other. Indeed, the authors themselves note that “specific dietary assessment” impacted the strength of the associations observed (abstract and conclusion). This is discrepancy is concerning and seemingly limits one’s ability to discern what is being observed in this analysis.
Given the small sample sizes, was there power enough to examine associations between individual ethnic groups and dairy intake? It seems that only Caucasians vs. Non-Caucasians had the power necessary for those types of analyses.
Lines 131-133: More details (or refences) should be provided on how the authors went from 24-hr/FFQ intakes to total dairy servings, especially for the FFQ part.
Rationale or a reference should be provided for why analyses were normalized per 1000 kcal. Why wasn’t energy simply included as a confounding variable? This reviewer is not aware of other manuscripts who have used per 1000 kcal when looking a food intake and its association with either diseases or nutrient levels
The results section needs to be revised to improve its readability. The presentation of both the 24-hour and FFQ data made the manuscript difficult to read and comprehend. This was especially true when the authors presented the data for the individual dairy products. Thus this reviewer was unsure what the main outcomes and impact of this manuscript would be to the readership of the journal.
Figure 1 seems superfluous given that the data presented there could easily be included in Table 1.
Lines 67: Define NHANES
Line 213: Include “data not shown” or report the data on energy intake in the manuscript.
Lines 231-234: This sentence does not seem to be supported by the presented data, given that the 24-hr and FFQ reported different results that were significant. Plus, the authors themselves acknowledge that assessment method impacted the results so how can they be similar?
Discussion section: The public health importance of this work should be expanded (e.g., the impact of reduced dairy intake has on meeting nutrient needs) as well as what the current guidance is for those with LNP
Lines 475 – 497: This section should be labeled strengths/limitations.
Author Response
"Please see the attachment."

Reviewer 2 Report
My teaching to students has always been not to discuss trends. This tells me that the number of observations was too small and the study needs to be redone with more subjects. Significant differences (<0.05% probability the difference could have happened randomly) is what should be discussed. Social sciences may look at this differently. Cultural differences must be involved here. If yogurt isn't part of the culture it won't be eaten, etc. The data seems to show this. My recommendation is that this paper only discuss significant differences. Trends can be acknowledged as a basis for future research, but, not to draw conclusions and make public health recommendations. The statistics are what make the paper confusing. You could make your point in a paper half this long. This is intended to be positive criticism!
Author Response
Reviewer 2My teaching to students has always been not to discuss trends. This tells me that the number of
observations was too small and the study needs to be redone with more subjects. Significant
differences (<0.05% probability the difference could have happened randomly) is what should
be discussed. Social sciences may look at this differently. Cultural differences must be involved
here. If yogurt isn't part of the culture it won't be eaten, etc. The data seems to show this. My
recommendation is that this paper only discuss significant differences. Trends can be
acknowledged as a basis for future research, but, not to draw conclusions and make public
health recommendations. The statistics are what make the paper confusing. You could make your
point in a paper half this long. This is intended to be positive criticism!
We thank the reviewer for this comment. We have removed all lines relating to
descriptions of trends throughout the discussion and have focused on the significant findings.Lines 501-509 and 595-601 have been added to further discuss the role of culture/ethnic
background and dairy intake.

Reviewer 3 Report
This study used samples from a previously characterized American cohort to determine correlations between the rs4988235 single nucleotide polymorphism (SNP), which has previously been used as predictor for lactase persistence in people of European descent. The study looked at ethnicity since previous studies showed decreased in dairy intake in Blacks, non-Hispanic Whites and Mexican-Americans individuals compared to Whites. The authors used ASA24 and FFQ collected data to determine if a specific genotype was directly correlated with the lack of consumption of dairy food products.
The study used a predesigned TaqMan SNP probe to determine the LP genotype, which is adequate for the analysis. The manuscript is clearly written.
Clear genotyping differences were observed between ethnicities included in the study. This finding is not particularly novel. Likewise, genotype was not associated with absolute consumption or non-consumption status, but instead with amount of dairy consumption.
Overall this is an adequate, well written study that contributes to the understanding of LP in North American populations. Limitations of the study are addressed in the discussion; however, considering the wealth of new data from microbiome studies, the authors did not include a discussion of how different microbiotas can impact tolerance to lactose.
Author Response
"Please see the attachment."

Round 2
Reviewer 1 Report
This reviewer has no additional comments.
Reviewer 2 Report
A good revision!